# Activity Profile, Heart Rate, Technical Involvement, and Perceived Intensity and Fun in U13 Male and Female Team Handball Players: Effect of Game Format

**DOI:** 10.3390/sports7040090

**Published:** 2019-04-19

**Authors:** Mads Madsen, Georgios Ermidis, Vincenzo Rago, Kristoffer Surrow, Jeppe F. Vigh-Larsen, Morten B. Randers, Peter Krustrup, Malte N. Larsen

**Affiliations:** 1Department of Sports Science and Clinical Biomechanics, SDU Sport and Health Sciences (SHS), University of Southern Denmark, 5230 Odense, Denmark; mamadsen@health.sdu.dk (M.M.); germidis1990@gmail.com (G.E.); surrow@outlook.com (K.S.); jeppefoged@hotmail.com (J.F.V.-L.); mranders@health.sdu.dk (M.B.R.); pkrustrup@health.sdu.dk (P.K.); 2Department of Movement Sciences and Wellness, “Parthenope” University of Naples, 80133 Naples, Italy; 3Centre of Research, Education, Innovation and Intervention in Sport, University of Porto, 4200-450 Porto, Portugal; vincenzo.rago@live.com; 4Portugal Football School, Portuguese Football Federation, 2780-186 Oeiras, Portugal; 5Section for Sport Science, Department of Public Health, Aarhus University, 8000 Aarhus, Denmark; 6School of Sport Sciences, Faculty of Health Sciences, UiT The Arctic University of Norway, 9019 Tromsø, Norway; 7Sport and Health Sciences, University of Exeter, Exeter, EX1 2LU, UK

**Keywords:** Team handball, team sport, children, small-sided-games

## Abstract

The aim of the study was to compare the activity pattern, heart rate (HR), technical involvement, and subjective perceptions in U13 boys and girls playing team handball in five game formats. Activity pattern, heart rate (HR), technical involvement, perceived fun, and exertion were recorded from four girls teams (n = 24) and four boys teams (n = 24) played during a 1-day tournament consisting of five different game formats of 15-min duration: Medium court size, 4v4 (M4v4), 5v5 (M5v5), and 6v6 (M6v6), and large court size, 5v5 (L5v5) and 6v6 (L6v6). Girls covered more total distance (TD) and high-speed running (HSR, 13–17.9 km·h^−1^) on the large court compared to the medium court (*p* < 0.05; ES = 2.1–3.1 and 1.2–2.5, respectively). Boys covered more distance as HSR and sprinting on the large court compared to the medium court, but only more TD on the large court compared to the medium court with the same number of players, (*p* < 0.05; ES = 1.0–1.8, 1.0–1.8, and 1.1–1.8, respectively). Team handball for U13 boys and girls is a high-intensity activity irrespective of court size. Increasing the court size with a fixed number of players increased the total distance and HSR, whereas manipulating the number of players on a fixed court size appears to influence technical involvement.

## 1. Introduction

Team handball is a physically demanding, high-intensity team sport with a high number of anaerobic actions, such as sprinting, jumping, shooting, and tackling. These actions are repeated multiple times throughout the game interspersed by low to moderate intense activities, adding a considerable aerobic component to the game. This activity pattern results in an average heart rate above 80% of the maximal heart rate [1,2]. 

Adult team handball is usually played indoor on a 40 m × 20 m court with two teams of seven players, including a goalkeeper. In Denmark, youth team handball is generally played on smaller courts up to the age of 10, after which the game is played on a full court. One previous study in adults investigated the matter of court dimensions in team handball, which indicated that changing the court dimensions affected perceived exertion and activity pattern and HR on the players, but no studies of this type were found in children or adolescents [3]. To our knowledge, only one study has been conducted in youth handball, with the purpose of analysing the activity profile of elite adolescent team handball players during matches [1]. Studies from other team sports, especially football, show that changes in the pitch/court size and number of players can affect the physical and technical demands of the game and player involvement [4,5], while others only found impacts on technical aspects [6]. The use of small-sided games (SSG) is extensively investigated in other sports and the results reveal that the training type has large beneficial effects on VO_2_max, agility, and repeated sprint ability and moderate beneficial effect on 10- and 20-m sprint performance, jump height, and intermittent endurance [7]. 

For several reasons, it would be of interest to identify the game format associated with the greatest physical load, most fun, and the most technical efforts. For coaches, knowing which format produces the most technical efforts and/or greatest physical load would allow them to make training more specific. In terms of public health, recreational team handball is an important physical activity for many children and adolescents, and small-sided high-intensity intermittent games have been shown to have a significant effect on bone mineralisation and fitness [8,9]. Furthermore, research has shown that children cite “fun” as the most important factor for participating in organised sport, which highlights the importance of this factor for keeping children interested in being active [10] and might help team handball associations to maintain members for a longer period. 

Based on previous findings in a study of the game format in U13 football [4], we hypothesized that a larger court will increase the total distance and that fewer players on the court will increase the involvement of the players in terms of more shots and duels per player.

The primary aim of the present study was to describe and compare the activity profile, HR, technical involvement, and perceived fun and exertion in U13 boys and girls in five different game formats. The secondary aim was to compare the outcomes between genders in the different game formats. 

## 2. Materials and Methods

### 2.1. Experimental Design

Eight U13 teams played a 1-day tournament in which each team played five different game formats of a 15-min duration. The games were played on a medium-sized court (30 m × 20 m, M) and a large-sized court (20 × 40, L) with 4 to 6 players per team as follows: M4v4, M5v5, M6v6, L5v5, and L6v6. The games were played indoors in a random order with official rules and goal sizes (2 m × 3 m). Activity pattern, HR, technical involvement, and perceptual responses were recorded. 

Only those outfield players who played a full match were included in the data analysis. Nonetheless, goalkeepers were allowed to assist their outfield team-mates during the offensive phase of L6v6 games. 

### 2.2. Participants

A total of four U13 boys (n = 24) and four U13 girls (n = 24) Danish teams participated in the study. 

### 2.3. Activity Pattern

The activity pattern was recorded using a wearable device incorporating a 200-Hz accelerometer and gyroscope (Polar Team Pro system, Polar, Kempele, Finland), which was placed on the lower sternum using an elastic band. 

Total distance (TD), peak speed (V_peak_), number of sprints (>18 km·h^−1^), and distance covered at standing/walking (St/W), 0–2.9; jog, 3.0–7.9; medium speed running (MSR), 8.0–12.9; high speed running (HSR), 13.0–17.9; and sprinting, >18.0 km·h^−1^ were measured [11,12]. Moreover, accelerations (Acc) and decelerations (Dec) were expressed in the following zones as: Acc ≤ 1.4 (0.1–1.4 m·s^−2^), Acc 1.5–2.3 (1.5–2.3 m·s^−2^), and Acc > 2.3 (>2.3 m·s^−2^) and Dec ≥ −1.4 (−0.1–−1.4 m·s^−2^), Dec −1.5–−2.3 (−1.5–−2.3 m·s^−2^), and Dec < −2.3 (<−2.3 m·s^−2^) [11,12]. 

### 2.4. HR Monitoring

HR was recorded with a Polar Team Pro (POLAR, Polar Electro Oy, Kempele, Finland) at 1-s intervals during each game. The HR data were expressed as mean and peak HR attained for each game format. In addition, the HR data were distributed into the following intensity zones: <120, 120–160, 160–180, 180–200, and >200 bpm [4].

Activity profile and HR data were stored in the device and downloaded using the manufacturer’s software (POLAR, Software version 1.3.1, POLAR, Polar Electro Oy, Kempele, Finland).

### 2.5. Rating of Perceived Exertion and Fun

Immediately after the end of each match, the players’ perceived exertion and perceived fun during the game were rated using two visual analog scales (VAS) [13]. For physical exertion, the players placed a mark on a 17.4-cm line ranging from ‘maximally demanding’ to ‘not demanding at all’, while for perceived fun, a similar line was used, ranging from ‘maximal fun’ to ‘not fun at all’. The result was obtained by measuring with a ruler the length (cm) from 0 to the mark made by the player.

### 2.6. Technical Variables

Shots on goal, number of goals scored, and number of duels were counted during the game by officials from the Danish handball federation. Each official followed one or two players per game. Afterwards, the percentage of successful shots was calculated. These technical variables were chosen since they were considered as the most important by the authors. Number of shots indicated the possibilities to create chances. Number of goals indicates how difficult it was to score (scoring changes might be bigger with a certain number of players on a certain court size). Duels were counted since they are a central and physically demanding part of the game. 

### 2.7. Statistical Analyses

Considering the fact that the players differed with regard to the number of game formats in which they participated, a linear mixed model with unstructured covariance was used for the analysis of differences between game formats and between gender [14]. The game format was set as the fixed effects and individual subjects and team were set as random effects. Physical, physiological, and perceptual variables were dependent variables. If a significant effect was found, multiple comparisons between game formats were analysed using a Bonferroni correction. Differences between game formats and genders were interpreted using effect sizes (ES) according to Hopkins and Marshall [15] as: Trivial (ES < 0.2), small (ES = 0.2–0.6), moderate (ES = 0.6–1.2), large (ES = 1.2–2.0), very large (ES = 2.0–4.0), and huge (ES > 4.0). When 90% confidence intervals (CI) overlapped positive and negative values, the effect was deemed to be unclear. Otherwise, the effect was deemed to be the observed magnitude [16]. Significance was set at *p* ≤ 0.05. Data analysis was performed using Statistical Package for Social Science statistical software (version 23, IBM SPSS Statistics, Chicago, IL, USA).

## 3. Results

### 3.1. Main Effect—Game Format in U13 Boys

#### 3.1.1. Activity Profile

The boys covered significantly more TD in L5v5 compared to M4v4 and M5v5 (*p* < 0.05; ES = 1.1 [0.5;1.7], 1.8 [1.2;2.5], respectively) and in L6v6 compared to M5v5 and M6v6 (*p* < 0.05; ES = 1.2 [0.6;1.8], 1.1 [0.6;1.6], respectively) (Table 1). In general, significantly more distance was covered by HSR and sprinting in L5v5 and L6v6 compared to M4v4 (*p* < 0.05; ES = 1.2 [0.6;1.8], 1.0 [0.4;1.6], and 1.2 [0.6;1.8], 1.0 [0.4;1.6], respectively), M5v5 (*p* < 0.05; ES = 1.8 [1.1;2.4], 1.1 [0.5;1.7], and 1.8 [1.2;2.5], 1.1 [0.5;1.7], respectively), and M6v6 (*p* < 0.05; ES = 1.2 [0.7;1.8], 1.3 [0.7;1.9], and 1.2 [0.7;1.8], 1.3 [0.7;1.8], respectively) (Figure 1). Furthermore, significantly more distance was covered by MSR in L5v5 and L6v6 compared to M5v5 (*p* < 0.05; ES = 2.1 [1.4;2.8] and 2.3 [0.4;2.9], respectively), M6v6 (*p* < 0.05; ES = 2.3 [1.6;2.9] and 1.3 [0.7;1.8], respectively), and in L5v5 compared to M4v4 (*p* < 0.05; ES = 1.4 [0.7;2.0]). Significantly less distance was covered by jogging in L6v6 compared to M4V4 (*p* < 0.05; ES = 1.0 [0.4;1.6]) (Figure 1).

The number of sprints was significantly greater in L5v5 and L6v6 compared to M5v5 (*p* < 0.05; ES = 0.9 [0.3;1.5] and 1.0 [0.4;1.5], respectively) and M6v6 (*p* < 0.05; ES = 1.0 [0.4;1.5] and 1.0 [0.5;1.5], respectively). V_peak_ was significantly higher in L5v5 compared to M6v6 (*p* < 0.05; ES = 1.1 [0.6;1.7]) and in L6v6 compared to M5v5 (*p* < 0.05; ES = 0.9 [0.4;1.5]) and M6v6 (*p* < 0.05; ES = 1.3 [0.7;1.8]) (Table 1).

L6v6 showed a significantly lower number of total accelerations compared to M4v4 and M6v6 (*p* < 0.05; ES = 1.2 [0.6;1.7] and ES = 1.1 [0.5;1.6]) (Table 1). Significantly fewer accelerations < 1.5 m·s^−2^ were measured in L5v5 compared to M6v6 (*p* < 0.05; ES = 1.0 [0.4;1.6]) and in L6v6 compared to M4v4, M5v5, and M6v6 (*p* < 0.05. ES = 1.4 [0.8;2.0], ES = 1.1 [0.6;1.7], and ES = 1.5 [0.9;2.0], respectively).

#### 3.1.2. Heart Rate

L5v5 showed significantly less time than M6v6 in the HR zone of 120–160 bpm (*p* < 0.05. ES = 0.8 [0.3;1.4]) (Figure 2).

#### 3.1.3. Technical Variables

Boys had more shots per player in M4v4 compared to M6v6 and L6v6 (*p* < 0.05; ES = 1.2 [0.8;1.4] and ES = 1.3 [0.7;1.9]) and scored more goals per player in M4v4 compared to L6v6 (*p* < 0.05; ES = 1.4 [0.8;2.0] (Table 2). 

#### 3.1.4. Subjective Ratings

Boys perceived L5v5 to be more physically demanding than M5v5 (*p* < 0.05; ES = 0.9 [0.3;1.4]) and M6v6 (*p* < 0.05; ES = 1.0 [0.5;1.6]). No differences between game formats were found in the boys’ rating of perceived fun during the games (Table 1). 

### 3.2. Main Effect—Game Format in U13 Girls

#### 3.2.1. Activity Profile

TD covered was significantly greater in L5v5 compared to M4v4, M5v5, and M6v6 (*p* < 0.05; ES = 2.2 [1.5;2.9], 3.1 [2.3;3.9], and 2.6 [3.3;1.9], respectively) and in L6v6 compared to M4v4, M5v5, and M6v6 (*p* < 0.05; ES = 2.1 [1.4;2.8], ES = 3.1 [2.3;3.9], and 2.5 [1.8;3.1], respectively) (Table 1). In L5v5 and L6v6, significantly more distance was covered by MSR and HSR compared to M4v4 (*p* < 0.05; ES = 1.8 [1.2;2.5], 2.9 [2.1;3.8], and 1.2 [0.6;1.8], 2.3 [1.6;3.0], respectively), M5v5 (*p* < 0.05; ES = 2.3 [1.6;3.1], 3.3 [2.4;4.1], and 1.8 [1.2;2.4], 2.5 [1.8;3.2], respectively), and M6v6 (*p* < 0.05; ES = 1.9 [1.3;2.5], 3.2 [2.4;4.0], and 1.2 [0.6;1.7], 2.5 [1.9;3.2], respectively), but L5v5 also covered significantly more distance by MSR than L6v6 (*p* < 0.05; ES = 1.0 [0.5;1.6]). L6v6 showed significantly more distance covered by sprinting compared to M5v5 (*p* < 0.05; ES = 1.1 [0.5;1.6]) and M6v6 (*p* < 0.05; ES = 1.1 [0.6;1.7]), and L5v5 showed significantly less distance covered by walking than M4v4 (*p* < 0.05; ES = 1.2 [0.6;1.2]) and M6v6 (*p* < 0.05; ES = 1.1 [0.5;1.6]) (Figure 1). The number of sprints was significantly greater in L5v5 compared to M6v6 (*p* < 0.05; ES = 0.9 [0.3;1.4]). V_peak_ was significantly higher in L5v5 and L6v6 compared to M5v5 and M6v6 (*p* < 0.05; ES = 1.1 [0.5;1.7], ES = 1.1 [0.5;1.7], and ES = 0.9 [0.4;1.5], ES = 1.0 [0.45;1.5], respectively) (Table 1). 

#### 3.2.2. Heart Rate

The girls had a significantly greater HR_mean_ in L5v5 compared to M5v5 (*p* < 0.05; E = 1.2 [0.6;1.8]). No other differences in HR_mean_ and HR_peak_ were found for the girls (Table 1).

#### 3.2.3. Technical Variables

Girls had more shots per player in M4v4 compared to M5v5 (*p* < 0.05; ES = 0.8 [0.2;1.4]), M6v6 (*p* < 0.05; ES = 1.1 [0.6;1.7]), L5v5 (*p* < 0.05; ES = 1.1 [0.5;1.7]), and L6v6 (*p* < 0.05; ES = 1.7 [1.2;2.3], and scored more goals per player in M4v4 compared to M5v5 (*p* < 0.05; ES = 0.9 [0.5;1.5]), M6v6 (*p* < 0.05; ES = 1.0 [0.4;1.6]), L5v5 (*p* < 0.05; ES = 1.2 [0.6;1.8]), and L6v6 (*p* < 0.05; ES = 1.4 [0.8;1.4]. 

#### 3.2.4. Subjective Ratings

Girls perceived L5v5 and L6v6 to be more physically demanding than M6v6 (*p* < 0.05; ES = 1.1 [0.6;1.7] and ES = 1.0 [0.5;1.5]). No differences were found in the girls’ rating of perceived fun during the games (Table 1). 

### 3.3. Main Effect—Gender

#### 3.3.1. Activity Profile

Boys covered largely greater TD than girls in all game formats played on medium-sized courts (*p* < 0.05; ES = 1.4 to 1.7). In all game formats, boys had a higher peak speed (*p* < 0.05; ES = 0.8 to 1.2) and a higher number of sprints *(p* < 0.05; ES = 0.9 to 1.4) than girls (Figure 3A). 

Boys covered a moderately greater distance by jogging compared to girls in M5v5, M6v6, and L5v5 (*p* < 0.05; ES = 0.6 to 0.8). Boys covered a moderately greater distance by MSR compared to girls in M4v4, M5v5, M6v6, and L6v6 (*p* < 0.05; ES = 0.6 to 1.2). Boys covered a moderately to largely greater distance by HSR compared to girls in M4v4, M5v5, and M6v6 (*p* < 0.05; ES = 0.8 to 1.7), but girls covered a moderately greater distance by HSR than boys in L6v6 (*p* < 0.05; ES = 0.6 [0.1;1.1]). Boys also covered a moderately to largely greater sprinting distance compared to girls in M4v4, M5v5, M6v6, and L6v6 (*p* < 0.05; ES = 0.7 to 1.3) (Figure 3B).

The boys performed a moderately higher number of total accelerations in M6v6 compared to the girls (*p* < 0.05; ES = 0.8 [0.3;1.4]) and a moderately higher number of total decelerations in M4v4, M5v5, and M6v6 (*p* < 0.05; ES = 0.7 to 0.8) (Figure 3A).

A greater moderate number of accelerations between 1.5 and 2.29 m·s^−2^ was found when boys played M4v4, M5v5, and L6v6 compared to girls (*p* < 0.05; ES = 0.8 to 1.1). Furthermore, a moderately to largely higher number of accelerations over 2.29 m·s^−2^ were found for boys playing M4v4, M5v5, and M6v6 compared to girls (*p* < 0.05; ES = 1.0 to 1.39) (Figure 3C). The boys had a moderately higher number of decelerations between 2.29 and 1.5 m·s^−2^ in M6v6 compared to girls (*p* < 0.05; ES = 1.0 [0.4;1.5]). Boys also had a moderately higher number of accelerations over 2.3 m·s^−2^ compared to girls in M4v4, M5v5, M6v6, and L5v5 (*p* < 0.05; ES = 1.1 to 1.2) (Figure 3C).

#### 3.3.2. Heart Rate

Girls had higher HR_peak_ than boys in L5v5 (*p* < 0.05; ES = 0.8 [0.3;1.4]). Girls also had more time in HR_zone1_ in L5v5 than boys (*p* < 0.05; ES = 0.7 [0.1;1.3]), but boys spent more time in HR_zone2_ in L6v6 compared to girls (*p* < 0.05; ES = 0.9 [0.4;1.4]) (Figure 3D)

#### 3.3.3. Technical Variables

Boys had more shots in M6v6 (*p* < 0.05; ES = 0.5 [0.5;0.6]) than girls and scored more goals in M5v5 (*p* < 0.05; ES = 0.5 [0.4;0.5]) than girls. In contrast, the rate of successful shots was higher for girls in M4v4 (*p* < 0.05; ES = 0.5 [0.0;0.5]) than for boys, but boys had more successful shots in M6v6 and L6v6 (*p* < 0.05; ES = 0.4 to 0.6) than girls (Figure 3E). 

#### 3.3.4. Subjective ratings

Girls perceived M6v6 and L5v5 to be more physically demanding compared to boys (*p* < 0.05; ES = 0.8 to 1.0). Boys perceived M4v4, M5v5, and L6v6 to be more fun compared to girls (*p* < 0.05; ES = 0.6 to 0.8) (Figure 3A).

## 4. Discussion

The main findings of the present study were that HR values were high, similar to other SSG with several number of sprints, accelerations, and decelerations during team handball games in U13 players irrespective of gender and game format. For girls, HR was even higher when playing L5v5 compared to M5v5. Both genders covered a greater TD when playing on the large-sized court compared to the medium-sized court with same number of players. The boys had a greater number of sprints when playing on the large-sized court than when playing on the medium-sized court with the same number of players. In general, differences in activity patterns were only seen when comparing the two court sizes, but not when the number of players was altered on the same-sized court.

The physiological response to exercise (mean HR) was high with HRs from 167 to 184 bpm in all game formats for both genders. These values are very similar to previous studies in adolescent team handball players [1] and U13 football players in small-sided games [4], and a little higher compared to school children playing small-sided games [9]. 

High HR is important for the health profile of children as well as adults, since aerobic high-intensity training has been shown to be more effective than continuous aerobic training in respect to improving cardiorespiratory fitness [17,18], which is associated with lower future body mass index body fat percentage, and metabolic syndrome [19]. In adults, two studies have shown beneficial health effects of team handball for inactive men and women [20,21], but no studies have investigated the health effects of recreational team handball in children or adolescents. Other types of small-sided ball games, including unihockey, football, and basketball, studied in a school setting in 8 to 10-year-old children, improved performance in the Yo-Yo Intermittent Recovery level 1 for children, bone mineral density, and structural and functional cardiac adaptations [8,9,22], so it can be expected that team handball could result in similar effects.

It might be claimed that the mean HR in the present team handball games are only a moderate-intensity activity, but team handball is an intermittent game with many actions of high intensity, as indicated by the HR_peak_ values between 192 and 202 bpm. Furthermore, team handball is a game with many changes of direction and in the present study, a little less than 400 total accelerations and decelerations were counted during each 15-min. game. To our knowledge, this is the first study to count accelerations during team handball. 

Most of the TD was covered by jogging for both genders irrespective of the court size, but when games were played on the large-sized court, the boys’ running intensity was higher and significantly more distance was covered by MSR, HSR, and sprinting. Thus, a change in court size, but not the number of players, had a large effect on the boys’ V_peak_ during games (ES = 1.3 to 2.3). This was also observed by Condivo et al. (2014) in adult men playing team handball, where the same number of players on a larger court covered a greater distance in higher speed zones compared to a smaller court. 

Regarding gender-related differences, boys covered more distance by sprinting compared to girls (ES = 0.7 to 1.3) in all game formats except L5v5 and had higher V_peak_ and a higher number of sprints in all game formats. This underlines the fact that boys are generally faster than girls [23,24]. In support of this observation, a study by Michalsik and Aagaard (2014) comparing adult elite women and male team handball players during matches found that men produced more high-speed running than women [25]. Furthermore, the study by Michalsik and Aagaard (2014) observed that female players covered greater TD than male players during a team handball game. Interestingly, the opposite was found in this study, but only on the medium-sized court, with boys covering greater TD than girls. No differences were found between genders on the large-sized court. The difference in TD between genders in this study may be explained by differences in the game pattern or simply the boys’ ability to run faster, since a greater distance was covered by MSR, HSR, and sprinting in the boys’ game. Moreover, the higher number of sprints in the same game formats might explain the greater number of accelerations and decelerations for boys, especially for accelerations over 2.3 m·s^−2^ on a medium-sized court and decelerations under −2.3 m·s^−2^ on a medium-sized court and in L5v5. 

Girls had more accelerations above 2.3 m·s^−2^ when playing on the large-sized court compared to the medium-sized court with the same number of players. This might be an effect of more space per player on the larger court, since the findings are similar to results from a study by Randers et al. (2018), who observed increased time in the two highest player load zones when the participants played 3v3 basketball on a full court compared to a half court [26]. 

In respect of the technical aspects of the game, girls had more shots and scored more goals per player when playing M4v4 compared to all other game formats. Boys had more shots per player when playing M4v4 compared to M6v6 and L6v6, but more goals were scored only in M4v4 compared to L6v6. This observation may simply be explained by the fewer number of players and greater space per player, which gave more opportunities to shoot. It is in line with studies of small-sided football games with different numbers of players, which showed a higher number of actions and more ball contact per player when the number of players decreased [4,6].

Boys reported it was more physically demanding playing L5v5 compared to M5v5 and M6v6, whereas girls found it more demanding playing both large-court formats compared to M6v6. For both genders, the observed differences in perceptual responses might be due to the larger court resulting in more TD covered and more time spent in higher speed zones, even though this did not result in increased HR. A positive relationship exists in the literature between total distance and rate of perceived exhaustion (RPE), and other studies have reported similar observations, with increased court size reported to elicit a greater response in RPE in team handball and football without any changes in HR [3,5,27]. In addition, boys reported it was more demanding playing M5v5 and L6v6 compared to girls. The difference might be due to a higher number of sprints in boys’ games, but this is not supported by differences in HR. In respect of RPE, it is important to stress that not only is RPE mediated by physiological factors, but psychological factors are also important, which may explain the difference in RPE in the present study [28].

A limitation of this study is that the maximal HR of the participants was unknown. A measure of maximal HR would have made it possible to calculate a precise estimate of the individual aerobic load during the games. Furthermore, precise individual measures of anthropometrics, state of matureness, and fitness components, such as maximal sprinting speed, would give a more precise description of the participants and might explain some of the differences between boys and girls, and give the opportunity to adjust the speed zones to gender. Gender differences have been studied in football [29], but have not been studied so far in team handball. 

Only a limited number of technical variables have been included in the study due to a limited number of officials. Other technical variables, such as placement of the shots and the shooters’ position on the court, would be of interest to investigate if a specific game format would favour shooting from specific positions or if more goals were scored in a specific part of the goal. 

Finally, it has not been possible to find any independent scientific validation studies for the equipment used to measure activity patterns. For this reason, the results of these measurements must be used with caution until the equipment is validated. 

## 5. Conclusions

HR was high with several sprints, accelerations, and decelerations in U13 team handball irrespective of gender and game format. Greater TD was covered when the court size was increased, and boys performed more sprints on the large-sized court compared to the medium-sized court with the same number of players. In the games with the fewest number of players on the court, there were more shots per player compared with the games with the most players on the court. 

## 6. Practical Implications

This study is the first to describe in detail and compare activity patterns, HR, subjective perceptions, and technical involvement by different game formats in U13 team handball. The various game types provide valuable information to coaches who wants to focus on specific technical or physical elements of the game in a game context and for team handball associations who want to adapt the game format to this age-group. 

## Figures and Tables

**Figure 1 sports-07-00090-f001:**
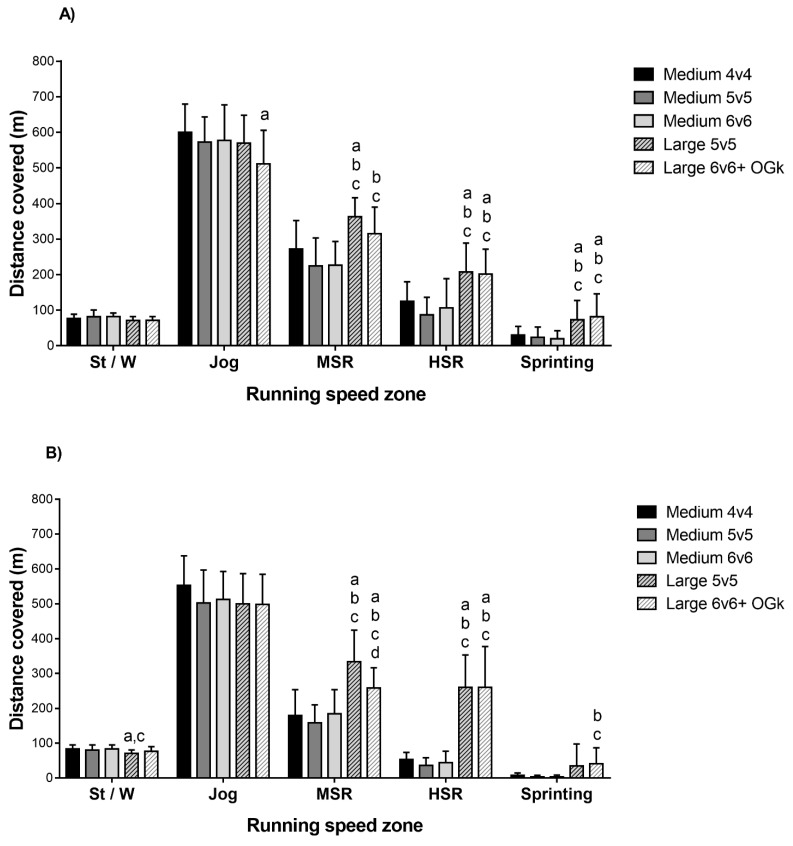
St/W, standing/walking; MSR, moderate-speed running; HSR, high-speed running. Distance covered for boys (**A**) and girls (**B**) in various speed zones. “a” denotes significantly different to Medium 4v4, “b” to Medium 5v5, “c” to Medium 6v6, “d” to Large 5v5 (*p* < 0.05).

**Figure 2 sports-07-00090-f002:**
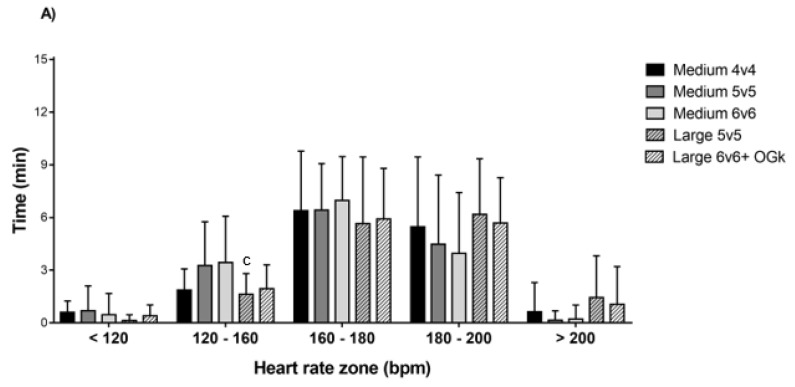
Time spent (min.) in different HR zones for boys (**A**) and girls (**B**). “a” denotes significantly different to Medium 4v4, “b” to Medium 5v5, “c” to Medium 6v6, “d” to Large 5v5 (*p* < 0.05). HR, heart rate.

**Figure 3 sports-07-00090-f003:**
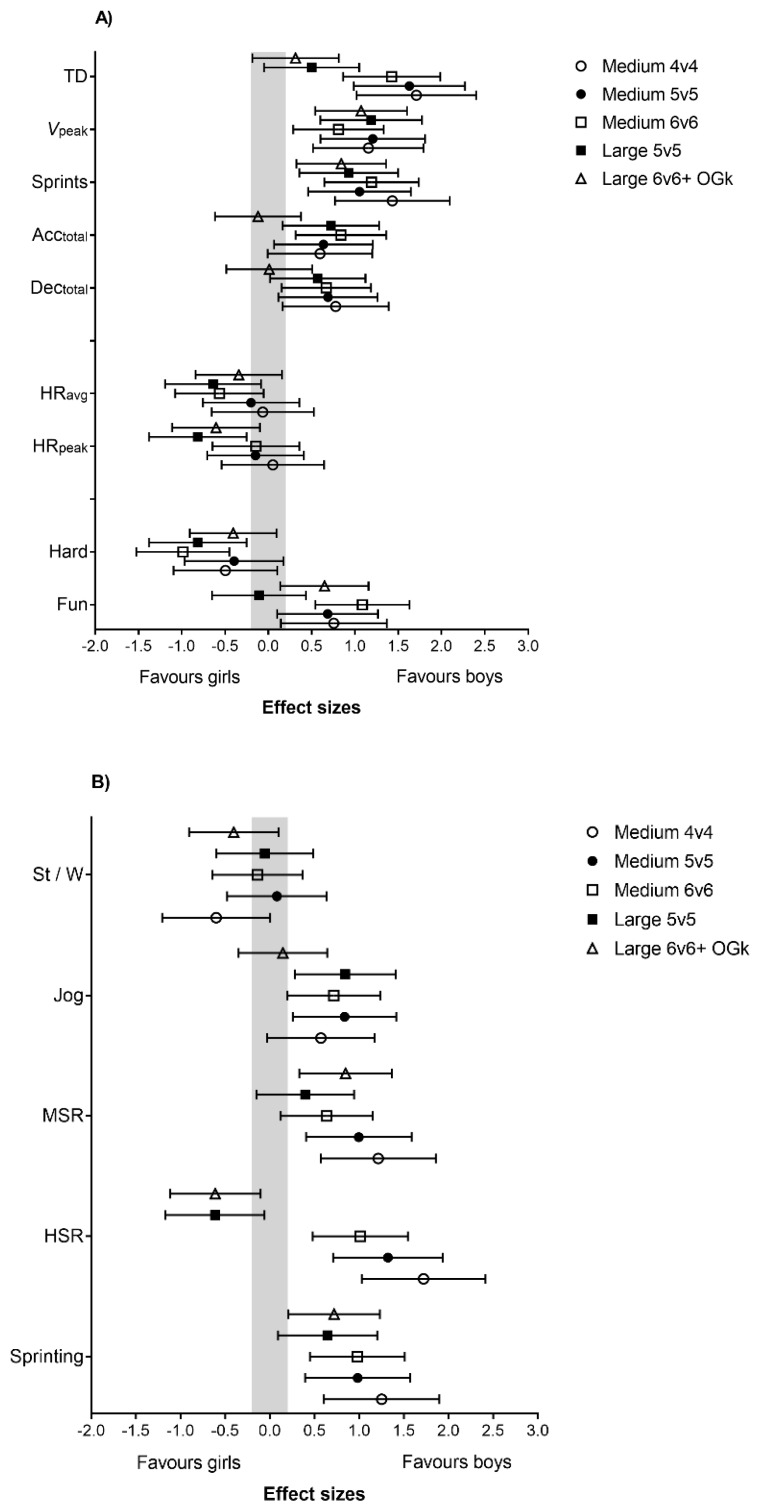
Differences in (**A**) overall physical and physiological demands, (**B**) activity profile, (**C**) accelerations and decelerations, (**D**) heart rate, and (**E**) technical demands between boys and girls during different team handball games. Effect size between boys and girls (CI 90%). CI, confidence intervals.

**Table 1 sports-07-00090-t001:** Activity pattern, heart rate and subjective perceptions for the five game formats.

	Gender	Medium 4v4	Medium 5v5	Medium 6v6	Large 5v5	Large 6v6 + OGk
*Activity pattern*						
Total distance (m)	Boys	1104 ± 156	990 ± 163	1014 ± 149	1284 ± 162 ^a,b^	1180 ± 158 ^b,c^
	Girls	877 ± 104	780 ± 77	828 ± 109	1201 ± 172 ^a,b,c^	1134 ± 136 ^a,b,c^
Peak speed (km·h^−1^)	Boys	23.0 ± 2.5	21.5 ± 2.9	20.7 ± 2.6	23.8 ± 2.8 ^c^	23.9 ± 2.4 ^b,c^
	Girls	20.2 ± 2.2	18.2 ± 2.5	18.5 ± 2.9	20.8 ± 2.2 ^b,c^	21.2 ± 2.7 ^b,c^
Sprints (counts)	Boys	5.5 ± 3.6	3.8 ± 3.6	3.8 ± 3.6	7.8 ± 5.1 ^b,c^	8.1 ± 4.8 ^b,c^
	Girls	1.7 ± 1.1	0.9 ± 1.1	0.8 ± 0.7	3.4 ± 4.4 c	4.3 ± 4.0
Total accelerations (counts)	Boys	200 ± 18	197 ± 28	200 ± 22	188 ± 16	177 ± 21 ^a,c^
	Girls	190 ± 16	179 ± 27	182 ± 21	173 ± 22	180 ± 21
Total decelerations (counts)	Boys	198 ± 20	192 ± 25	199 ± 21	195 ± 18	185 ± 21
	Girls	185 ± 12	175 ± 26	184 ± 23	182 ± 25	184 ± 22
*Heart rate*						
Mean heart rate (bpm)	Boys	173.8 ± 10.1	167.2 ± 13.3	167.2 ± 11.8	177.1 ± 10.5	174.9 ± 9.9
	Girls	174.5 ± 10.2	169.8 ± 12.1	173.5 ± 10.3	184.3 ± 11.8 ^b^	179.4 ± 15.8
Peak heart rate (bpm)	Boys	195.4 ± 11.7	191.5 ± 9.7	191.6 ± 9.7	195.8 ± 8.0	196.9 ± 7.9
	Girls	194.9 ± 7.5	192.9 ± 9.8	193.0 ± 10.1	202.1 ± 7.4	202.6 ± 10.7
*Subjective perceptions*						
Hard (au)	Boys	7.0 ± 4.0	8.9 ± 4.0	9.4 ± 3.8	6.0 ± 2.7 ^b,c^	7.4 ± 1.9
	Girls	9.2 ± 4.9	10.7 ± 4.8	12.8 ± 3.1	8.9 ± 4.0 ^c^	9.0 ± 4.4 ^c^
Fun (au)	Boys	8.4 ± 4.1	7.5 ± 4.4	7.4 ± 4.1	5.1 ± 2.8	5.5 ± 2.4
	Girls	5.2 ± 4.4	4.8 ± 3.0	3.6 ± 2.8	5.5 ± 3.1	3.7 ± 3.0

^a^ denotes significantly different to Medium 4v4, ^b^ to Medium 5v5, ^c^ to Medium 6v6, ^d^ to Large 5v5 (*p* < 0.05).

**Table 2 sports-07-00090-t002:** Tecnical variables for the five game formats.

	Gender	Medium 4v4	Medium 5v5	Medium 6v6	Large 5v5	Large 6v6 + OGk
Shots (counts)	Boys	5.0 ± 2.4	3.4 ± 1.9	3.2 ± 2.3 ^a^	3.3 ± 1.6	2.3 ± 1.8 ^a^
	Girls	6.1 ± 3.4	3.9 ± 2.0 ^a^	3.1 ± 2.1 ^a^	3.2 ± 1.9 ^a^	2.0 ± 1.5 ^a^
Goals (counts)	Boys	2.5 ± 1.1	1.8 ± 1.3	1.7 ± 1.3 ^a^	1.8 ± 1.2	1.0 ± 1.1 ^a^
	Girls	3.8 ± 2.8	1.7 ± 1.7 ^a^	1.6 ± 1.7 ^a^	1.3 ± 1.1 ^a^	0.9 ± 1.0 ^a^
Successful shots (%)	Boys	56.2 ± 24.4	51.3 ± 34.2	43.9 ± 38.4	61.7 ± 29.9	38.3 ± 38.4
	Girls	57.4 ± 33.8	44.3 ± 34.6	42.8 ± 36.9	36.8 ± 30.0	36.4 ± 37.4
Duels (counts)	Boys	5.3 ± 4.5	4.0 ± 4.0	3.8 ± 3.7	3.3 ± 2.0	3.5 ± 2.6
	Girls	2.0 ± 1.9	2.1 ± 2.1	1.4 ± 1.6	1.7 ± 1.6	1.8 ± 2.3

^a^ denotes significantly different to Medium 4v4, ^b^ to Medium 5v5, ^c^ to Medium 6v6, ^d^ to Large 5v5 (*p* < 0.05).

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
