# Peer review of "Activity Profile, Heart Rate, Technical Involvement, and Perceived Intensity and Fun in U13 Male and Female Team Handball Players: Effect of Game Format"

_sports, 2019, doi:10.3390/sports7040090_

Round 1

Reviewer 1 Report

General

It is a nice and comprehensive study. Well done. I have added some comments to strengthen the paper.

The back and forth between performance variables for coaches and then the public health information is a little confusing. Who is the intended target audience? Given there is more of a focus on the performance, and the Journal, I would suggest removing the public health comments.

There is a large body of work on SSG in football, maybe some comparisons across sports could add some perspective.

Limitations, and practical implications for coaches are required.

Introduction

Introduction is fine, maybe consider my comments on the public health sections, and replace with SSG findings across sports such as football to provide some context.

Methods

The variable chosen are accepted and explained.

Was there Ethical approval for the study?

Define VAS scales

Why were the technical variables of shots and goals chosen? What about location of shots, assists, etc? What about defensive variables?

I think whatever Technical variables you choose you need to explain why the variables were chosen, and why they are of importance in Handball.

Please explain why the Bonferonni post hoc test was chosen.

Results

There are a lot of results, and there is clearly a lot of work to present. Maybe try to condense where possible.

Lines 177 – 178. M 6 v 6 listed twice.

Table descriptions should go above the table.

Discussion

First sentence of the Discussion. The variables are high…. Compared to what?

Line 327. Could girls having more goals scored be due to the location of shots? As per my comment in the Methods.

Line 357. Same as point above. High compared to what?

There are occasions in the Discussion where you make statements without references, please check these. Line 283-284 is an example.

Author Response

General

It is a nice and comprehensive study. Well done. I have added some comments to strengthen the paper.

The back and forth between performance variables for coaches and then the public health information is a little confusing. Who is the intended target audience? Given there is more of a focus on the performance, and the Journal, I would suggest removing the public health comments.

Thank you for your comment. The target group is primarily sport and health scientists and coaches. We agree that the comments about health can be confusing in a performance perspective, but we believe that organizing sports for children with high intensity is important for public health, since we see that participating in different sports are associated with the likelihood of living up to the WHO recommendations (Hebert), and that it does matter, which sports you participate in.

*There is a large body of work on SSG in football, maybe some comparisons across sports could add some perspective.

Thank you for you comment. We have added a section to the introduction highlighting the benefits of SSG:

The use of small-sided games is well investigated in other sports and the results reveals that the training type has large beneficial effect on VO2max, agility and repeated sprint ability and moderate beneficial effect on 10- and 20-m sprint performance, jump height and intermittent endurance [7]. (Line 58)

Limitations, and practical implications for coaches are required.

Thank you for your comment. We have added following to the limitations og the technical variables:

Only a limited number of technical variables have been included in the study due to a limited number of officials. Other technical variables such as placement of the shots and the shooters position on the court would be of interest to investigate if a specific game format would favor shooting from specific positions or if more goals were scored in a specific part of the goal. (Line 372)   

A section of practical implications is added to the end of the paper:

Practical implications

This study is the first to detailed describe and compare activity pattern, HR, subjective perceptions and technical involvement by different game formats in U13 team handball. The various game types provide valuable information to coaches who wants to focus on specific technical or physical elements of the game in a game context and for team handball associations who wants to adapt the game format to the age-group. (Line 390)

Introduction

Introduction is fine, maybe consider my comments on the public health sections, and replace with SSG findings across sports such as football to provide some context.

Thank you for your comment. A new section is added as described earlier.

Methods

The variable chosen are accepted and explained.

Was there Ethical approval for the study?

Thank you for your comment. The study is following the local Danish ethical guidelines. A study like this should not be reported, because of the type of measurements and since it is not an intervention. 

Define VAS scales

Visual analog scale are now defined in the text.

Why were the technical variables of shots and goals chosen? What about location of shots, assists, etc? What about defensive variables?

I think whatever Technical variables you choose you need to explain why the variables were chosen, and why they are of importance in Handball.

Thank you for your comment, we agree that these needs to clearly explained. Following is added to the method section:

These technical variables were chosen since they were considered as the most important by the authors. Number of shots indicated the possibilities to create chances. Number of goals indicates how difficult it was to score (scoring changes might be bigger with a certain number of players on a certain court size). Duels were counted since they are a central and physical demanding part of the game. (Line 120)

Please explain why the Bonferonni post hoc test was chosen.

The Bonferroni’s test controls the Type I error rate very well. Bonferroni has more power when the

number of comparisons is small.

Field, A. (2009). Discovering statistics using SPSS, Third Edition. Sage Publications.

Results

There are a lot of results, and there is clearly a lot of work to present. Maybe try to condense where possible.

Lines 177 – 178. M 6 v 6 listed twice.

Thank you for noticing. It is now corrected. 

Table descriptions should go above the table.

Thank you for noticing. It is now corrected. 

Discussion

First sentence of the Discussion. The variables are high…. Compared to what?

Thank you for your comment. We agree that “high” by itself might be to unclear. The sentence is now changed so HR is “high” similar to other SSG, were the description “high” is used. The similarities are further described later in the discussion.

Line 327. Could girls having more goals scored be due to the location of shots? As per my comment in the Methods.

Unfortunately, we haven’t tracked the location of the shots. Of cause, well placed shots will more often end as a goal. More space per player might give better conditions (e.g. less pressure on the shooter, shorter distance to goal) for the shooters since more goals are scored, but this is not investigated in the study. We have added the few technical variables to the limitation of the study.

Line 357. Same as point above. High compared to what?

See above.

There are occasions in the Discussion where you make statements without references, please check these. Line 283-284 is an example.

Thank you for your comment. We agree that some statements have unclear references, as the given example. The text has now been reviewed and references have been added and in some cases the sentences are changed to cite the reference more correctly. 

Reviewer 2 Report

First, it is very important that the authors arranged the article according to the template journal.Also, the keywords.

Lines 47-51 - It is not necessary to present how the handball is played, where they play, length and width of the terrain...etc....this thing he know anyone, please delete this sentence.

Please include in Introduction chapter current research on this topic, also which is the novelty of the research.

Lines 114-115 - Please detailed how a adult official had the time to count the shots on goals, number of goals, of duels....etc?????

Line 180 - Please include the title of table.

Also, please include the conclusions of the research.

Author Response

First, it is very important that the authors arranged the article according to the template journal.Also, the keywords.

Thank you for your comment. The arrangement has been reviewed, and changes have been made according to the journals template. 

Lines 47-51 - It is not necessary to present how the handball is played, where they play, length and width of the terrain...etc....this thing he know anyone, please delete this sentence.

Thank you for you comment. We have deleted most of the sentence, but we have kept some of if to state the difference between adult and youth team handball. (Line 47)

Please include in Introduction chapter current research on this topic, also which is the novelty of the research.

Thank you for your comment. Little research has been done on the topic and is now described in the introduction:

One previous study in adults have investigated the matter of court dimensions in team hand-ball, which indicated that changing the court dimensions affected perceived exertion and activity pattern and HR on the players, but no studies of this type were found in children or adolescents [3]. To our knowledge, only one study has been conducted in youth handball, with the purpose to analyses the activity profile of elite adolescent team handball players during matches [1]. (Line 49)

Lines 114-115 - Please detailed how a adult official had the time to count the shots on goals, number of goals, of duels....etc?????

Thank you for your comment. We agree that it haven’t been clear enough how this part was conducted. Each official followed 1-2 player per game and counted the described variables. This have been added to the method section. 

Shots on goal, number of goals scored, and number of duels were counted during the game by officials from the Danish handball federation. Each official followed one or two players per game. (Line 118)

Line 180 - Please include the title of table.

Thank you for noticing. Title is now added.

Also, please include the conclusions of the research.

Thank you for you comment. The article has now been updated with a conclusion.

Reviewer 3 Report

Line 23: 11 and 12-year-old….

(same on line 70)

Line 25: lowercase medium and large

Line 53: confused by what you mean “is not further adapted”.  Also you probably don’t need the word different (since you identify them as older)

Line 69: including hypotheses would be of benefit

Line 78: “with a different number of players” seems awkward

Maybe: The games (M4v4, M5v5…) were played on …

Or have a separate sentence for the #s

Table 1 and 2– might be better to superscript a, b, c

Line 233 – should it be 0.7

Line 238 – should it be 0.8

Line 258 – should it be 0.8

Author Response

Comments and Suggestions for Authors

Line 23: 11 and 12-year-old….

(same on line 70)

Thank you for noticing. We have decided to be consistent using U13 instead. 

Line 25: lowercase medium and large

Thank you for noticing. The changes have been made.

Line 53: confused by what you mean “is not further adapted”.  Also you probably don’t need the word different (since you identify them as older)

Thank you for the comment. Then sentences have now been changes to be clearer: “In Denmark, youth team handball the game is generally played on smaller courts up to the age of 10, after which the game will be played on a full court.” (Line 48)

Line 69: including hypotheses would be of benefit

Thank you for your comment. We have added this hypothesis to the end of the introduction:

Based on previous findings in a study of game format in U13 football [4], we hypothesis that a larger court will increase the total distance and that fewer players on the court will increase the involvement of the players in terms of more shots and duels per player. (Line 71)

Line 78: “with a different number of players” seems awkward

Maybe: The games (M4v4, M5v5…) were played on …

Or have a separate sentence for the #s

Thank you for your comment. We agree that sentence was awkward. The sentence is now changed to following:

The games were played on a medium-sized court (30x20 m, M) and a large-sized court (20x40, L) with 4 to 6 players per team as follows: M4v4, M5v5, M6v6, L5v5 and L6v6. (Line 83)

Table 1 and 2– might be better to superscript a, b, c

Thank you for noticing. The changes have been made.

Line 233 – should it be 0.7

Line 238 – should it be 0.8

Line 258 – should it be 0.8

Thank you for noticing. The changes have been made.

Round 2

Reviewer 2 Report

-